# Genome-wide analysis of lncRNA stability in human

**Kaiwen Shi**[1], **Tao Liu**[1], **Hanjiang Fu**[2], **Wuju Li**[1]*, **Xiaofei Zheng**[2]*

**1** Institute of Military Cognition and Brain Sciences, Academy of Military Medicine, Beijing, China, **2** Beijing Key Laboratory for Radiobiology, Beijing Institute of Radiation Medicine, Beijing, China

* wujuchina@126.com (WL); xfzheng100@126.com (XZ)

**Data Availability Statement:** All supplementary datasets can be downloaded from our webpage http://ccb1.bmi.ac.cn:081/lncrnastability/.

**Funding:** This research was supported by grants from National Natural Science Foundation of China (No.91540202 for XZ, and No.31271404 and

## Abstract

Transcript stability is associated with many biological processes, and the factors affecting mRNA stability have been extensively studied. However, little is known about the features related to human long noncoding RNA (lncRNA) stability. By inhibiting transcription and collecting samples in 10 time points, genome-wide RNA-seq studies was performed in human lung adenocarcinoma cells (A549) and RNA half-life datasets were constructed. The following observations were obtained. First, the half-life distributions of both lncRNAs and messanger RNAs (mRNAs) with one exon (lnc-human1 and m-human1) were significantly different from those of both lncRNAs and mRNAs with more than one exon (lnc-human2 and m-human2). Furthermore, some factors such as full-length transcript secondary structures played a contrary role in lnc-human1 and m-human2. Second, through the half-life comparisons of nucleus- and cytoplasm-specific and common lncRNAs and mRNAs, lncRNAs (mRNAs) in the nucleus were found to be less stable than those in the cytoplasm, which was derived from transcripts themselves rather than cellular location. Third, kmers-based protein−RNA or RNA−RNA interactions promoted lncRNA stability from lnc-human1 and decreased mRNA stability from m-human2 with high probability. Finally, through applying deep learning−based regression, a non-linear relationship was found to exist between the half-lives of lncRNAs (mRNAs) and related factors. The present study established lncRNA and mRNA half-life regulation networks in the A549 cell line and shed new light on the degradation behaviors of both lncRNAs and mRNAs.

## Author summary

Transcript stability is important for many biological processes. However, little is known about the features related to human lncRNA stability. Through quantitative analysis between the half-lives of lncRNAs (mRNAs) and various factors, we found a nonlinear relationship between the half-lives of lncRNAs (mRNAs) and the related factors and their combinations. Our research provided a comprehensive understanding of lncRNA stability. Further efforts are needed to develop an accurate quantitative prediction model for the half-lives of lncRNA (mRNA).

31471244 for WL). The funders had no role in study design, data collection and analysis, decision to publish, or preparation of the manuscript.

**Competing interests:** The authors have declared that no competing interests exist.

## Introduction

Many studies have shown that transcript stability has a close relationship with their biological function [1–6], and plays a vital role in determining the transcript level and sensing environmental changes [7–15]. For example, with the Iron (Fe) deficiency, the *Saccharomyces cerevisiae* protein Cth2 specifically downregulated many messenger RNAs (mRNAs) through binding AU-rich elements, which encoded proteins associated with many Fe-dependent processes [11]. Additionally, a close link exists between mRNA half-life regulation and breast cancer [16]. To understand the biological processes deeply, elucidating the factors involved in transcript stability is necessary.

Up to the present, many factors associated with mRNA stability have been studied, which included transcript length [4,6,8,17,18], GC contents [2,4,18–20], RNA secondary structures [6,8,20], microRNA (miRNA)-mRNA interactions [2], cellular locations [19,21], protein-RNA interactions [22], and optimal codon contents in mRNAs [1,3,7,20,23–28]. For example, there existed a significant negative correlation between half-lives of mRNAs and their lengths in human and *Escherichia coli* [17]. In *Arabidopsis thaliana* [2], Narsai et al. showed that miRNA targets often have short half-lives, and mRNAs with more than one exon often have longer half-lives. Additionally, some studies showed that there was a close relationship between codon usages and half-lives of mRNA in yeasts and human. However, all aforementioned studies mainly focused on mRNA stability analysis. Only a few studies are associated with noncoding RNAs (ncRNAs) stability [3,19,21].

NcRNAs are a class of transcripts with little or no potential encoding proteins, which can be classified into short RNAs such as miRNAs and piRNAs [29,30], and long noncoding RNAs (lncRNAs) with their lengths more than 200 nt [31]. These ncRNAs play an important regulatory role in many biological processes. For example, miRNA can downregulate their target mRNAs through miRNA-mRNA interactions [32]. Additionally, lncRNAs are involved in many biological processes such as transcription, translation, RNA modification, and epigenetic modification of chromatin structures [33,34]. Furthermore, a close relationship exists between ncRNAs functions and their stability [3]. Therefore, a systematic investigation of ncRNA stability is of importance.

In their study, Tani et al. [3] determined the half-lives of 11,052 mRNAs and 1418 ncRNAs in HeLa cells, with an average 6.90 h and 7.0 h, respectively. Through classifying ncRNAs into stable and unstable groups with half-lives 4 h as the threshold, they found that ncRNAs with housekeeping functions were enriched in the stable group, which included tRNAs, snoRNAs, and small Cajal body-specific RNAs. In their study, Clark et al. [19] obtained the half-lives of 11,773 mRNAs and 823 lncRNAs in mouse, with an average (median) 7.7 h (5.1 h) and 4.8 h (3.5 h), respectively. They also systematically investigated lncRNA stability in terms of genomic positions, cellular location, GC contents, or exon number, and found that lncRNAs enriched in the nucleus were less stable than those in the cytoplasm, and lncRNAs with more than one exon often have longer half-lives. In their study, Ayupe et al. [21] applied custom microarray to determine the half-lives of 791 intronic, 695 antisense lncRNAs, and 4204 mRNAs in Hela cells, with the medians 3.9 h, 2.1 h, and 3.2 h, respectively. They also found that lncRNAs enriched in the nucleus were unstable than the remaining lncRNAs. Even so, there are some issues unhandled for lncRNA stability analysis. For example, is there any relationship between half-lives of lncRNAs and their RNA secondary structures or miRNA-lncRNA interactions or protein-lncRNA interactions? and is there any difference in stability-regulating mechanisms for both lncRNAs and mRNAs? Furthermore, according to the database NONCODEv5 [31], there are 172,216 human lncRNAs and 131,697 mouse lncRNAs,

respectively. From the perspective of statistics, a large number of lncRNAs with half-lives is necessary to investigate the relationship between half-lives of lncRNAs and related factors. To this end, we first applied RNA-seq to determine the degradation profiles of transcripts and calculated the half-lives of transcripts genome-widely. Then we comprehensively studied the relationship between lncRNA stability and potential factors in A549 cell lines.

## Results and discussion

### Calculation of half-life expression profile

We obtained the half-lives for 34,268 lncRNAs and 33,029 mRNAs, respectively by sampling strategy (see Methods and Fig 1). Considering the biological uncertainty and no annotations for novel transcripts, we only took into account the transcripts with annotations and their half-lives less than 50 h and finally obtained the half-life datasets including 33,285 lncRNAs (lnc-human) and 24,710 mRNAs (m-human) for further analysis (S3 and S4 Tables). Their cumulative distributions are provided in Fig 2, from which we found that about 80% half-lives of lncRNAs were less than 5 h. However, only about 60% mRNAs were present with half-lives less than 5 h. The average half-llife of lncRNAs and mRNAs was 3.96 h and 6.35 h, and the median was 2.76 h and 4.18 h, respectively. The Kolmogorov-Smirnov test demonstrated a significant difference between the two populations with a *P* value of 0. Therefore, the lncRNAs are less stable than mRNAs, which agrees with lncRNA stability analysis in mice [19] and humans [35]. Furthermore, the coefficient of variation of both lncRNA and mRNA was 1.13 and 1.04, respectively, indicating that half-lives of lncRNAs had a wide variation. Additionally, we applied the GO annotation tool (http://geneontology.org/) to annotate the biological processes of mRNAs with their half-lives less than 5 h. The annotation results showed that the enriched GO terms were mainly associated with the regulation of biosynthetic process (FDR = 7.85E-12), regulation of cellular biosynthetic process (FDR = 1.31E-11), regulation of cellular metabolic process (FDR = 1.32E-11), and so on. However, The GO annotation for mRNAs with their half-lives more than 5 h indicated that the enriched GO terms were primarily related to the metabolic process (FDR = 1.01E-44), organic substance metabolic process (FDR = 1.65E-39), primary metabolic process (FDR = 1.62E-38), and so on. These results were similar to the previous conclusions [2–4,10,19,36], demonstrating that our half-life datasets are

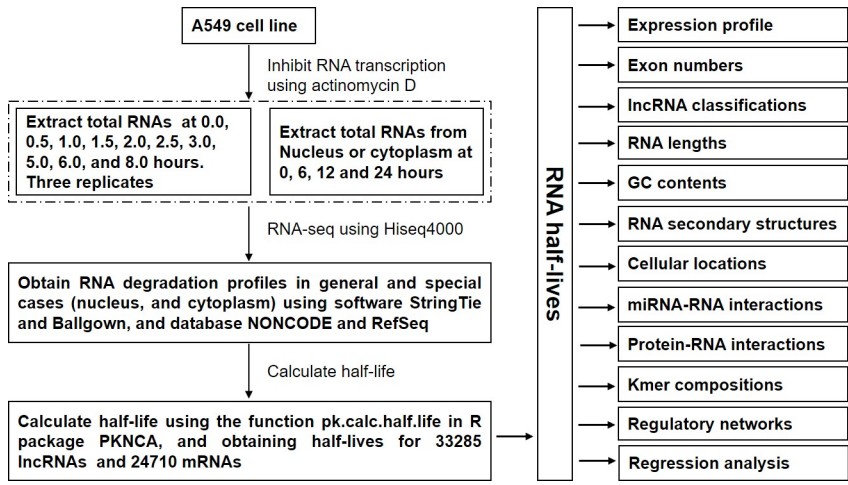

**Fig 1. Flowchat for the whole experiments and data analysis.**

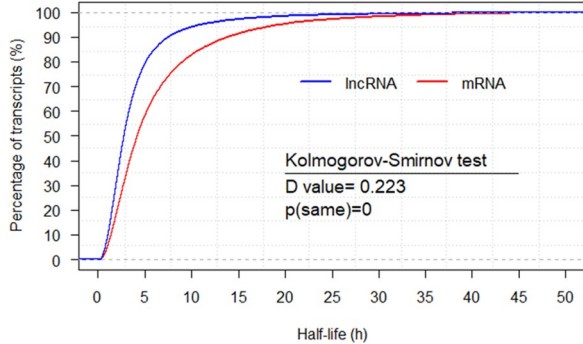

**Fig 2. Half-life cumulative distributions for both lncRNAs and mRNAs (h).**

comparable to the previous ones. Furthermore, from the annotation of mRNAs with their half-lives less than 5 h, we deduced that most lncRNAs play regulatory roles.

## The relationship between half-lives of lncRNAs and their expression levels

Some published studies demonstrated the relationship between the half-lives of transcripts and their expression levels [1,10,21,23,37,38]. For example, the study on *E. coli* showed that there was no relationship between mRNA stability and their expression levels [1]. However, the Ayupe et al. study showed a positive correlation between half-lives of lncRNAs and their expression levels [21]. Additionally, the Spearman correlation from the Clark dataset [19], which included half-lives of 823 ncRNA and 11,773 mRNA and related expression profiles, showed a significant positive correlation for mRNAs and no relationship with ncRNAs. The Spearman correlation was calculated using the datasets from the present study, and the results indicated a significant positive correlation with a *P* value 2.30E-319 for lncRNAs, and 0.0 for mRNAs. Therefore, from the perspective of statistics, the transcripts with a higher expression level mean that they are generated quickly and imply that they decay slowly and have longer half-lives.

## Exon number-based lncRNA stability analysis

The relationship between the half-lives of transcripts and the number of exons present in them was studied in *A. thaliana* and mice [2,19]. In *A. thaliana*, mRNA transcripts with more than one exon were significantly more stable than those with only one exon. However, such a relationship for lncRNAs has not been explored. In mice, both lncRNAs and mRNAs with more than one exon were more stable than those with only one exon. To check whether this relationship existed in human, we first extracted the number of exons in each lncRNA from the NONCODE database (S5 Table) [31]. Among 33,285 lncRNAs with half-lives, 12,465 lncRNAs had 1 exon (lnc-human1), and 20,820 lncRNAs had more than 1 exon (lnc-human2). Subsequently, we calculated the Spearman correlation between half-lives of lncRNAs and the number of exons present in them. The results indicated a significant negative correlation (*P* = 3.49E-21), indicating that lncRNAs with fewer exons tended to be more stable. In fact, the average half-life for lnc-human1 and lnc-human2 was 4.16 h and 3.85 h, respectively. The Kolmogorov-Smirnov test showed a significant difference between the two populations with a *P* value of 0 (Fig 3). The exon number-based lncRNA stability results in humans are different from those in mice [2,19].

We also checked the relationship in mRNAs. Among 24,710 mRNAs with half-lives, 474 mRNAs had 1 exon (m-human1) and 24,236 mRNAs had more than 1 exon (m-human2, S6

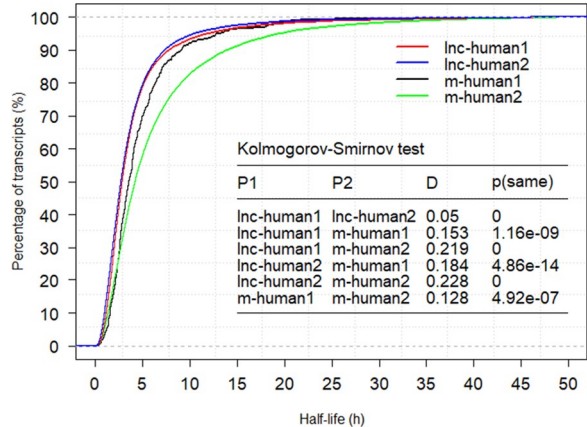

**Fig 3. Half-life cumulative distributions for lnc-human1, lnc-human2, m-human1, and m-human2 (h).**

Table). A weak positive correlation was observed between half-lives of mRNAs and the number of exons present in them ($P = 7.72E-3$), indicating that more exons in mRNAs would enhance their stability. In fact, the average half-life for m-human1 and m-human2 was 4.81 and 6.38 h, respectively. Therefore, our results in humans were consistent with the results from *A. thaliana* and mice [2,19]. The Kolmogorov-Smirnov test indicated a significant difference between the two populations ($P = 4.92E-7$, Fig 3). Fig 3 shows a significant difference between the population lnc-human1 and m-human1 ($P = 1.16E-9$), lnc-human1 and m-human2 ($P = 0$), lnc-human2 and m-human1 ($P = 4.86E-14$), and lnc-human2 and m-human2 ($P = 0$). Therefore, for stability analysis, it is necessary to divide the lncRNA and mRNA into different populations using the information of exon number.

## LncRNA classification-based lncRNA stability analysis

According to the genomic positions of lncRNA genes, lncRNAs can be classified into the following types: sense, intergenic, intronic, antisense, and divergent [31] (see Methods). Here we considered only the first four classes because of their clear definition [39]. A total of 5518 sense, 17,095 intergenic, 4602 intronic, and 5319 antisense lncRNAs (S7 Table) were present, with their average half-live of 4.49 h, 3.88 h, 3.75 h and 3.75 h, respectively. The one-way analysis of variance (ANOVA) analysis indicated a significant difference among four classes ($P = 1.59E-22$). Additionally, the Kolmogorov-Smirnov test indicated a significant difference between class sense and class intergenic ($P = 1.11E-16$), or class intronic ($P = 3.33E-16$), or class antisense ($P = 1.90E-10$) (Fig 4). Further analysis demonstrated that the average half-life for the four classes (sense, antisense, intergenic, and intronic) from lnc-human1 was 5.64 h, 3.94 h, 4.08 h, and 3.89 h, which were larger than 4.32 h, 3.69 h, 3.75 h, and 3.50 h from lnc-human2, respectively. Therefore, For each class, the average half-life from lnc-human1 is always larger than that from lnc-human2, and class sense is the most stable class among the our classes in either lnc-human1 or lnc-human2.

## Relationship between half-lives of lncRNAs and their lengths or GC contents or secondary structures

Although the relationship between the half-lives of transcripts and their lengths, GC contents, or secondary structures have been extensively studied [1,2,4,6,8,10,17–20,23,26], there is no clear relationship at present. For example, some studies showed no correlation between half-

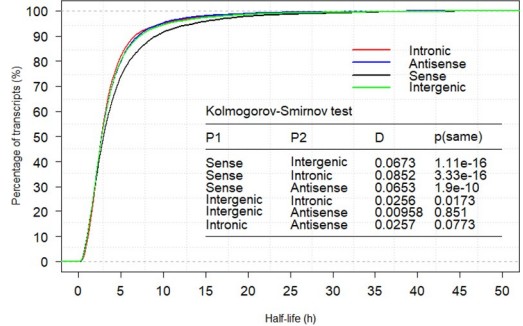

**Fig 4. Half-life cumulative distributions for class sense, intergenic, intronic and antisense lncRNAs (h).**

life and mRNA length [1,10,23,26]. However, a study demonstrated the negative correlation between half-lives of mRNAs and their lengths in human and *E. coli* [17]. Additionally, the present studies on the relationship between the half-lives of transcripts and their secondary structures mainly focused on mRNAs. For example, it has been shown that mRNA secondary structures are not predictive of their half-lives in *E. coli* and *Bacillus subtilis* [1,40]. In yeast, a positive relationship was demonstrated between the secondary structure free energy of mRNA 5' UTRs, CDSs, or 3' UTRs and their half-lives [20]. However, a study reported a negative relationship between mRNA 3' UTR secondary structures and their half-lives [6]. Up to the present, no systematic investigation has been reported in humans. To this end, we considered the RNA secondary structures-based analysis of transcripts stability. Here we only considered the transcripts with their lengths less than 10,000 nt. There are 32080 lncRNAs and 24211 mRNAs with their secondary structures predicted (S8–S10 Tables). Furthermore, since we do not know which regions of transcript 5' (3') ends are the best regions for studying the relationship, we first extracted a series of sequence fragments from transcripts 5' (3') ends. For lncRNAs, we extracted fragments of lengths 50, 60, . . ., and 300 from 5' or 3' end, respectively. For mRNAs, we extracted the fragments flanking the initial codon and termination codon with the lengths 100, 120, . . ., and 600, respectively. Therefore, we extracted 26 fragments for each transcript 5' end or 3' end. Then we applied the RNAFOLD program to predict second structures of all sequences [41] (S11–S14 Tables). Finally, we applied Spearman correlation analysis to study the relationship on the population lnc-human, lnc-human1, lnc-human2, m-human, m-human1, and m-human2, separately. The detailed results are provided in Fig 5, from which we obtained the following observations.

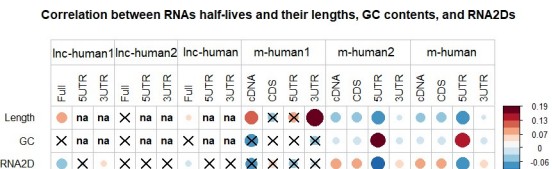

**Fig 5. Spearman correlation between the half-lives of transcripts and their lengths, GC contents, or secondary structures (RNA2D) were displayed for lnc-human1, lnc-human2, lnc-human, m-human1, m-human2, and m-human, respectively.** For lncRNA, the full, 5UTR, and 3UTR stand for full lengths, 5' UTR and 3' UTR local fragments. For mRNAs, the cDNA, CDS, 5UTR and 3UTR stands for cDNAs, CDSs, 5' UTR fragments, and 3' UTR fragments, in which the 5UTR and 3UTR represent the regions with the most significant *P* value. The signs "×" and "na" stand for no statistical significance at *P* = 0.01 and missing value, respectively, and the sizes of round dots stand for the correlation coefficients.

First, a significant positive correlation existed between the half-lives of lncRNAs and their lengths in lnc-human1 ($P$ = 1.90E-19) and no relationship existed in lnc-human2 ($P$ = 1.89E-1). Therefore, the weak positive correlation in the lnc-human ($P$ = 1.02E-4) was mainly contributed from lnc-human1. Additionally, we also checked which lncRNA classes had the aforementioned correlation. A total of 714 sense, 6905 intergenic, 1295 antisense, and 2974 intronic lncRNAs were obtained from lnc-human1, with the average half-life 5.64 h, 4.08 h, 3.94 h, and 3.89 h, respectively. The results indicated a significant positive correlation between half-lives of lncRNAs and their lengths for class intergenic ($P$ = 2.64E-11) or class intronic ($P$ = 3.26E-12), and no significant relationship for class sense ($P$ = 1.21E-1) or class antisense ($P$ = 6.86E-3). For mRNAs, a significant negative correlation existed between half-lives of mRNAs and their lengths of cDNAs ($P$ = 1.69E-26), CDSs ($P$ = 1.22E-25), 5' end UTRs ($P$ = 2.14E-72), and 3' end UTRs ($P$ = 2.86E-4) in m-human2. In m-human1, a weak positive correlation existed between half-lives of mRNAs and their lengths of cDNAs ($P$ = 7.84E-3) and 3' end UTRs ($P$ = 6.72E-5). Additionally, we also observed that mRNA 3' end UTR lengths played a contrary role in m-human1 and m-human2. The overall significant negative correlations between half-lives of mRNAs and their lengths of cDNAs ($P$ = 2.26E-23), CDSs ($P$ = 6.67E-25), 5' end UTRs ($P$ = 2.59E-69), and 3' end UTRs ($P$ = 2.94E-3) were mainly contributed from m-population2. Finally, we also checked the relationship in the mouse dataset, which was downloaded directly from their webpage [19]. A total of 434 lncRNAs (lnc-mouse1) and 440 mRNAs (m-mouse1) had 1 exon, and 389 lncRNAs (lnc-mouse2) and 11,333 mRNAs (m-mouse2) had more than 1 exon. The results demonstrated no significant correlation between half-lives of lncRNAs and their cDNA lengths in lnc-mouse1 ($P$ = 6.89E-1), lnc-mouse2 ($P$ = 8.62E-1), or the whole population ($P$ = 5.95E-1). Thus, the results from lnc-human1 and lnc-mouse1 were different. However, the reason for no relationship in lnc-mouse1 might be attributed to the fewer number of lncRNAs. To this end, we randomly extracted 434 lncRNAs from lnc-human1, which were the same number of lncRNAs in lnc-mouse1, and calculated the correlation between half-lives of lncRNAs and their lengths for 1000 times. The results indicated that there are about 500 times with their $P$ values more than 0.1 Therefore, although a significant positive correlation existed between the half-lives of the lncRNAs and their lengths in lnc-human1 ($P$ = 1.90E-19), such a relationship could not be assured on the sample with only 434 lncRNAs. However, with the increase in the number of lncRNAs sampled from lnc-human1, the number of times providing $P$ > 0.1 gradually decreased (Fig 6). Fig 6 shows the number of times with their $P$ > 0.1 were about 2 or 0, respectively, when 3000 or 4000 lncRNAs were randomly

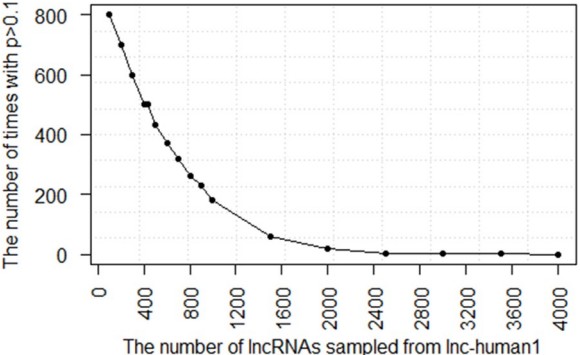

**Fig 6. Relationship between the number of lncRNAs sampled from lnc-human1 and the number of times with their $P$ > 0.1 in 1000 simulations, in which $P$ was calculated from the Spearman correlation analysis between the half-lives of lncRNAs and their lengths.**

extracted from lnc-human1 for 1000 times. Therefore, a large number of samples was necessary to deduce the reliable relationship between half-lives of lncRNAs and their lengths. For mRNAs, a significant negative correlation existed between half-lives of mRNAs and their cDNA lengths in m-mouse1 ($P$ = 7.12E-3), or m-mouse2 ($P$ = 2.30E-52), or the whole population ($P$ = 1.07E-36), which was basically the same as that in human.

Second, no significant correlation existed between half-lives of lncRNAs and their GC contents for both lnc-human1 ($P$ = 9.55E-1) and lnc-human2 ($P$ = 6.21E-1). For mRNAs a significant negative correlation existed between half-lives of mRNAs and their GC contents from cDNAs ($P$ = 1.99E-7), CDSs ($P$ = 3.86E-11), and 3' end UTRs ($P$ = 7.07E-7), and a strong positive correlation for 5' end UTRs ($P$ = 1.71E-131) in m-human2. However, no significant relationship was found in m-human1. Therefore, the overall relationships between the half-lives of mRNAs and their GC contents were mainly derived from m-human2. Furthermore, the GC contents of both 5' end UTRs and 3' end UTRs played a contrary role in regulating mRNA stability. The higher GC contents of 5' end UTRs or lower GC contents of 3' end UTR promote mRNA stability. Finally, according to the mouse dataset, no significant correlation existed in lnc-mouse1 ($P$ = 3.75E-1) and lnc-mouse2 ($P$ = 9.85E-2), which was consistent with the results from humans. For mRNAs, a negative correlation existed between half-lives of mRNAs and their GC contents for m-mouse1 ($P$ = 2.09E-5), and a positive correlation existed for m-mouse2 ($P$ = 2.62E-42) [19]. Therefore, the role of mRNA GC contents in regulating stability in humans is different from that in mice.

Third, a significant negative correlation existed between half-lives of lncRNAs and their full-length secondary structure free energy in lnc-human1 ($P$ = 8.24E-19), indicating that the lncRNAs with more stable secondary structures had longer half-lives. However, no relationship was found in lnc-human2. Therefore, the negative correlation between lncRNA secondary structures and their half-lives in lnc-human ($P$ = 2.40E-6) was mainly derived from lnc-human1. For mRNAs, a significant positive correlation existed between half-lives of mRNAs and their secondary structure free energy of cDNAs ($P$ = 3.74E-32), CDSs ($P$ = 3.21E-28), and 3' end ($P$ = 5.94E-13), and a strong negative relationship with 5' end secondary structure free energy ($P$ = 1.92E-80) in m-human2. The corresponding regions are − 70 ~ 70 nt for 5' end (flanking initial codon) and 300 ~ −300 nt for 3' end (flanking termination codon), respectively. Therefore, stable mRNA 5' end secondary structure or unstable mRNA 3' end secondary structure promoted mRNA stability. However, such a relationship was not found in m-human1. Furthermore, full-length transcript secondary structures played a contrary role in lnc-human1 and m-human2.

The aforementioned results suggested that classifying the whole lncRNA (mRNA) into two populations using the information of exon numbers was necessary. Otherwise, the unique features in lnc-human1 and m-human2 could not be found.

## Cellular location-based lncRNA stability analysis

A few studies showed that lncRNAs were less stable in the nucleus than in the cytoplasm through enrichment analysis [19,21]. To investigate the relationship between the half-lives of transcripts and their cellular location (nucleus or cytoplasm) directly, we applied RNA-seq to determine the expression levels of transcripts separately in the nucleus and the cytoplasm at four time points, 0, 6, 12 and 24 h, after inhibiting transcription using actinomycin D, and obtained the expression profiles of 37,888 lncRNAs and 152,370 mRNAs (S15 and S16 Tables). Then we employed two strategies, enrichment analysis and half-life calculation, to study the influence of cellular location on transcripts stability.

For the first strategy, we defined a transcript enriched in the nucleus if its expression meets the following conditions: ① N-count > 0, C_count > 0, and N_fpkm/C_fpkm ≥ 2; or ② C_count = 0 and N_count ≥ 3, in which N_fpkm and N_count stand for the expression level and the number of fragments, respectively, from the transcript in the nucleus, and C_fpkm and C_count stand for the expression level and the number of fragments, respectively, from the transcript in the cytoplasm. Similarly, we defined a transcript enriched in the cytoplasm if its expression meets the following conditions: ① C_count > 0, N-count > 0, and C_fpkm/N_fpkm ≥ 2; or ② N_count = 0, and C_count ≥ 3. Therefore, the whole transcriptome is classified into three populations with population1 for the transcripts enriched in the nucleus, population2 for the transcripts enriched in the cytoplasm, and population3 for the remaining transcripts. Through the analysis of the intersection set with the half-life datasets from lnc-human and m-human, we obtained the half-life information for the transcripts in different populations (population1, population2, and population3) in different time points (Table 1) as follows.

First, for both lncRNAs and mRNAs at each time point, we obtained the following inequality: the average half-life in population1 < the average half-life in population3 < the average half-life in population2, indicating that lncRNA or mRNA stability in the nucleus is less than that of lncRNAs or mRNAs in the cytoplasm on average. Furthermore, the Kolmogorov-Smirnov test showed a significant difference between population1 and population2. One-way ANOVA also demonstrated that the half-lives had a significant difference among the three populations. Second, lncRNAs were mainly enriched in the nucleus during the degradation process. Table 1 shows that the ratio of the number of lncRNAs enriched in the nucleus to the number of lncRNAs enriched in the cytoplasm was always greater than 5. However, the ratio for mRNAs is around 1. Third, the number of lncRNAs or mRNAs in the cytoplasm gradually decreased with time from 0 to 6, 12, and 24 h, which indicated that some transcripts were degraded. In the nucleus, the number of lncRNAs had the same trend. But the number of mRNAs did not have this trend, showing that some new transcripts might have been generated.

The second strategy was to calculate the half-lives of transcripts directly using the aforementioned expression datasets and the function pk.calc.half.life in the PKNCA package. We finally obtained the half-lives of nucleus-specific 5758 lncRNAs and 13,093 mRNAs (S17–S18 Tables), cytoplasm-specific 1515 lncRNAs and 9833 mRNAs (S19–S20 Tables), and nucleus

**Table 1. Detailed information for both lncRNAs and mRNAs with half-lives enriched in the nucleus and the cytoplasm.**

| Type | Time (h) | Num (T) | Num (N) | Num (M) | Num (C) | Ratio (N/C) | $t_{1/2}$ (N) | $t_{1/2}$ (M) | $t_{1/2}$ (C) | $P_{ws.test}$ | $P_{AOV}$ |
|---|---|---|---|---|---|---|---|---|---|---|---|
| lncRNA | 0 | 17,361 | 11,728 | 3422 | 2211 | 5.30 | 3.92 | 4.06 | 4.73 | 0.00 | 5.09E-14 |
| | 6 | 17,480 | 10,165 | 5355 | 1960 | 5.19 | 3.75 | 4.29 | 4.85 | 0.00 | 1.05E-27 |
| | 12 | 17,236 | 10,157 | 5257 | 1822 | 5.57 | 3.81 | 4.17 | 5.12 | 0.00 | 1.00E-30 |
| | 24 | 16,247 | 8,835 | 5749 | 1663 | 5.31 | 3.83 | 4.29 | 4.97 | 0.00 | 5.07E-23 |
| mRNA | 0 | 19,469 | 6,472 | 5385 | 7612 | 0.85 | 6.10 | 6.80 | 7.31 | 0.00 | 3.97E-24 |
| | 6 | 19,717 | 5,808 | 7827 | 6082 | 0.95 | 5.36 | 6.83 | 7.91 | 0.00 | 1.69E-90 |
| | 12 | 19,608 | 6,416 | 7359 | 5833 | 1.10 | 5.15 | 7.11 | 8.10 | 0.00 | 1.40E-129 |
| | 24 | 19,231 | 6,782 | 7168 | 5281 | 1.28 | 5.07 | 7.36 | 8.34 | 0.00 | 4.07E-161 |

Time (h) column stands for four time points, namely 0, 6, 12, and 24 h, for enrichment analysis with the transcript inhibited at 0 h. For each time point. Num (T) stands for the total number of transcripts with half-lives, which is equal to the sum of Num (N) for the number of transcripts enriched in the nucleus, Num (M) for the number of transcripts in the middle state (not enriched in both the nucleus and the cytoplasm), and Num (C) for the number of transcripts enriched in the cytoplasm. Ratio (N/C) stans for the ratio of Num (N)/Num (C). $t_{1/2}$ (N), $t_{1/2}$ (M), and $t_{1/2}$ (C) stand for the average half-lives of transcripts enriched in the nucleus, middle state, and cytoplasm, respectively. $P_{ws.test}$ is the P value calculated from the Kolmogorov-Smirnov test between the transcripts enriched in the nucleus and the transcripts enriched in the cytoplasm. $P_{AOV}$ is the P value from the one-way ANOVA among the transcripts in the nucleus, middle state, and cytoplasm, respectively.

and cytoplasm-common 491 lncRNAs and 2496 mRNAs with their adjusted $R^2 \geq 0.7$ [1,21,23], respectively (S21–S24 Tables). Fig 7 shows the cumulative distribution of both lncRNAs and mRNAs, and the observations are as follows.

First, a significant difference was found between the nucleus- and cytoplasm-specific lncRNAs (mRNAs), with the $P$ value as 0 (0) from the Kolmogorov-Smirnov test (Fig 7A). The average half-life for lncRNAs (mRNAs) in the nucleus and the cytoplasm was 10.52 h (13.32 h) and 14.20 h (15.17 h), respectively. Second, however, no significant difference was found between the nucleus and cytoplasm-common lncRNAs (lncRNA: $P$ = 2.69E-2; mRNA: $P$ = 5.80E-1; Kolmogorov-Smirnov test, Fig 7B). The average half-life for lncRNAs (mRNAs) in the nucleus and the cytoplasm was 13.51 h (15.91 h) and 14.12 h (15.71 h), respectively. Therefore, the significant difference between the half-lives of nucleus- and cytoplasm-specific lncRNAs (mRNAs) might be attributed to the transcripts themselves or other factors rather than the cellular location (nucleus or cytoplasm). The GO annotation for the common 2496 mRNAs demonstrated that these mRNAs were mainly associated with the macromolecule metabolic process (FDR = 1.55E-54), cellular metabolic process (FDR = 4.84E-47), nitrogen compound metabolic process (FDR = 8.06E-46), and so on. Therefore, the functions of the common lncRNAs might be related to metabolic processes.

To detect the reason why there was a significant difference in half-life distribution between the populations of nucleic- and cytoplasm-specific lncRNAs (mRNAs), we calculated the difference in kmer distribution in the two populations using the Kolmogorov-Smirnov test, in which kmer length was assigned 1, 2, 3, 4, 5 and 6, respectively (see the following section "Kmer compositions-based lncRNA stability analysis"). We also considered the GC contents. The results indicated that among the 5461 features, 875 (3550) features had their FDR <1.0E-5 for lncRNAs (mRNAs). Furthermore, GC-type kmers, such as CG, CGG, and CCG, usually had less content in nucleic-specific lncRNAs (mRNAs) than in cytoplasm-specific lncRNAs (mRNAs). For example, the GC content for nucleic-specific lncRNAs (mRNAs) and cytoplasm-specific lncRNAs (mRNAs) was 4.46E-1 (4.64E-1) and 4.96E-1 (5.23E-1), respectively. These kmers with a significant difference in nucleus- and cytoplasm-specific lncRNAs (mRNAs) may be the parts of potential target regions for protein-RNA or RNA-RNA interactions, making these lncRNAs (mRNAs) less stable in the nucleus than in the cytoplasm.

## miRNA target prediction-based lncRNA stability analysis

MiRNAs play an important role in regulating gene expression. Through the interaction between miRNAs and their target mRNAs, miRNA targets are often degraded [42]. However,

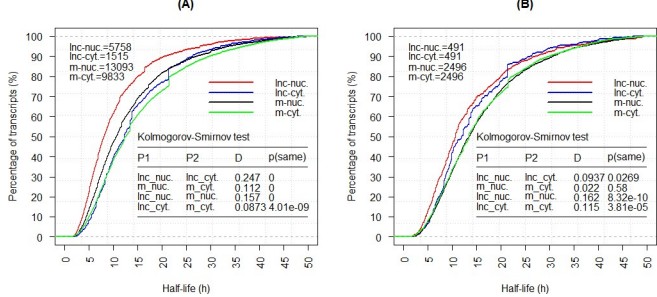

**Fig 7.** The Half-life cumulative distributions and the related Kolmogorov-Smirnor test for nucleus- and cytoplasm-specific lncRNAs and mRNAs (A), and nucleus and cytoplasm-common lncRNAs and mRNAs (B), in which the lnc-nuc., lnc-cyt., m-nuc., and m-cyt. stands for lncRNAs in the nucleus, lncRNAs in the cytoplasm, mRNAs in the nucleus, and mRNAs in the cytoplasm, respectively.

no clear relationship has been found yet between lncRNA stability and miRNA-lncRNA interactions. To systematically detect the association, we first applied the program miRanda to predict miRNA-lncRNA interactions [43], in which the full lncRNA sequences were taken as the potential targets. We obtained 6,197,809 miRNA-lncRNA interactions including 2656 miRNAs and 33,252 lncRNAs (S25 Table). Then, we calculated the number of miRNA-binding sites in each lncRNA. Finally, we applied a Spearman correlation analysis to check the association. The results indicated a significant positive correlation in lnc-human1 ($P$ = 9.31E-13), and no correlation in lnc-human2 ($P$ = 2.37E-2) or lnc-human ($P$ = 6.44E-2). Furthermore, through considering the distribution of binding sites of each miRNA in lnc-human1, we found 32/37 miRNAs significantly promoting lncRNA stability using the Kolmogorov-Smirnov test (FDR <0.01, S26 Table), in which the denominator 37 stands for the total number of miRNAs with their FDR <0.01. The top five miRNAs were hsa-miR-508-5p (FDR = 8.87E-6), has-miR-6747-3p (FDR = 1.28E-5), has-miR-3652 (FDR = 2.43E-5), hsa-miR-4433a-3p (FDR = 4.45E-5), and has-miR-504-3p (FDR = 4.58E-5), respectively. All these miRNAs promote lncRNA stability.

As a comparison, we also investigated the role of miRNAs in regulating mRNA stability. A total of 7,465,073 miRNA-mRNA interactions [43] were found, including 2654 miRNAs and 24,710 mRNAs (S27 Table). The results indicated a significant negative correlation between the half-lives of mRNAs and the number of potential miRNAs binding sites on the whole mRNA population m-human ($P$ = 2.86E-53). This finding demonstrated that miRNA-mRNA interactions reduce mRNA stability. However, the significant negative relationship only existed in m-human2 ($P$ = 2.09E-56) rather than in m-human1 ($P$ = 1.37E-1). We also checked which mRNA regions (5' end UTRs, CDSs, and 3' end UTRs) took part in regulation in m-human2. The results showed a significant negative correlation for 5' end UTR ($P$ = 3.87E-24), CDS ($P$ = 2.94E-50), and 3' end UTR ($P$ = 7.58E-12) at the same time. Further, the Kolmogorov-Smirnov test (FDR<0.01) indicated that 337 miRNAs related to 5' end UTRs, 254 miRNAs related to CDSs, and only 21 miRNAs related to 3' end UTRs (S28–S30 Tables). The total number of miRNAs associated with mRNA stability is 365. Therefore, only about 13.75% (365/2654) miRNAs closely associated with mRNA stability. All interactions associated with these miRNAs reduce mRNA stability. Furthermore, 5' UTR and CDSs of mRNAs are the main regions for miRNAs to regulate mRNA stability. Finally, the recent reports [44,45] revealed that both miR-7-5p and miR-141-3p were involved in regulating transferrin receptor-1 mRNA stability. However, according to the report from CORRAL et al. [46], neither miR-7-5p nor miR-141-3p is a major mediator of TfR1 mRNA degradation. We checked the relationship between these two miRNAs and half-lives of mRNAs in our datasets and found that the FDR value for these two miRNAs was 1.00. Therefore, from the perspective of statistics, these two miRNAs have little chance to be the regulator of TfR1 mRNA stability.

From the aforementioned results, we concluded that miRNAs play a contrary role in regulating the stability of both lncRNAs from lnc-human1 and mRNAs from m-human2. For lnc-human1, miRNA-lncRNA interactions extend the half-lives of lncRNAs with a big chance (32/37). However, for m-human2, the interactions between miRNAs and mRNAs promote mRNA degradation.

## Protein-lncRNA interaction-based stability analysis

Protein-RNA interactions play an important role in many biological processes such as RNA splicing, miRNA transport, and RNA stability [47]. To study the relationship between RNA-binding protein and RNA stability, we first downloaded the dataset of protein recognition RNA motifs from the database ATtRACT [48]. A total of 2297 unique motifs are associated

with human. The number of RNA-binding proteins is 179. Then, we calculated the occurrence numbers of each 2297 motifs across 33,285 lncRNAs with half-lives. For each lncRNA, the occurrence numbers of all motifs were added and considered as the ability of lncRNA binding to proteins. We found a positive correlation between half-lives of lncRNAs and their protein-binding ability on lnc-human1 ($P$ = 5.74E-19) and lnc-human ($P$ = 1.79E-4), and no relationship on lnc-human2 ($P$ = 1.44E-1). Further, the Kolmogorov-Smirnov test showed 319/331 motifs and 97/100 proteins promoting lncRNA stability in lnc-human1 (FDR<0.01, S31 and S32 Tables), in which the denominator 331 and 100 stand for the total number of motifs and proteins with their FDR <0.01. The top five motifs are UAGGA (FDR = 5.22E-10), CAGUGA (FDR = 6.72E-10), AGUAG (FDR = 7.72E-10), AUAAUU (FDR = 1.31E-9), and AGAGG (FDR = 1.64E-9). Additionally, the top five RNA-binding proteins were SF1 (FDR = 2.53E-10), RBMS3 (FDR = 2.44E-9), SRSF7 (FDR = 3.44E-9), ACO1 (FDR = 1.54E-8), and TRA2A (3.22E-8). Therefore, protein-lncRNA interactions mainly promoted lncRNA stability from lnc-human1 with a big chance (motif: 319/331; protein: 97/100).

For mRNAs, there existed a significant negative correlation between half-lives of mRNAs and their protein-binding ability on m-human2 ($P$ = 3.41E-26) and m-human ($P$ = 4.11E-23), and a weak positive correlation for m-human1 ($P$ = 8.20E-3). Therefore, protein-mRNA interactions mainly influence mRNA stability from m-human2. Further analysis showed that the half-lives had a significant negative correlation with the number of motifs in 5' end UTRs ($P$ = 2.86E-72), CDSs ($P$ = 2.08E-27), and 3' end UTRs ($P$ = 6.70E-4). The results from the Kolmogorov-Smirnov test (FDR < 0.01) indicated that 457/467 motifs and 131/138 proteins related to 5' end UTRs, 132/132 motifs and 15/16 proteins to CDSs, and 19/19 motifs and 2/2 proteins to 3' UTRs, which reduce mRNA stability (S33–S38 Tables). Therefore, mRNAs 5' end UTRs and CDSs are the main regions for RNA-binding protein to regulate mRNA stability.

In sum, the aforementioned analysis show that RNA-binding proteins play a contrary role in regulating lncRNA and mRNA stability. For lncRNAs, RNA-binding proteins mainly promote lncRNA stability from lnc-human1. However, RNA-binding proteins mainly reduce mRNA stability from m-human2.

## Kmer composition-based lncRNA stability analysis

The previous section demonstrated that protein recognition RNA motifs play an important role in regulating the stability of both lncRNAs from lnc-human1 and mRNA from m-human2. A more general case, that is kmers composition-based transcripts stability analysis, was considered in the present study. According to the database ATtRACT [48], RNA motif lengths varied from 4 to 12 nt with 7 nt as the median. However, with the kmer length becoming large, the number of all possible kmers became huge. For example, a total of 16,384 possible kmers existed for kmer = 7 ($4^7$ = 16,384), and the compositions for most kmers became zero. Therefore, we only considered the cases of kmer = 3, 4, 5, and 6. Finally, we calculated kmer compositions in two ways: (1) normal kmer compositions were calculated, in which any two neighbor kmers overlapped with the length of kmer-1 (K1); (2) kmers compositions were calculated in the same way as the codon content in mRNA sequences, in which any two neighbor kmers did not overlap (K2).

For K1, 1/1 (kmer = 3), 16/16 (kmer = 4), 41/41 (kmer = 5), and 158/158 (kmer = 6) kmers positively correlated with the half-lives of lncRNAs in lnc-human1 (FDR <0.01, S39–S42 Tables), in which the denominators stand for the total number of kmers with their FDR <0.01. The existence of these kmers promote lncRNA stability. The results are similar to the roles of protein recognition RNA motifs or miRNAs in lnc-human1. Therefore, these kmers might be the parts of potential motifs for protein-lncRNA interaction or miRNA-lncRNA interaction

regions. In lnc-human2, however, only 3/3 (kmer = 4) and 3/3 (kmer = 5) kmers negatively correlated with the half-lives of lncRNAs (FDR < 0.01), which indicate that these six kmers reduce lncRNA stability (S43 and S44 Tables). As a comparison, we also considered the Spearman correlation in m-human1 and m-human2, respectively. For mRNA CDS-based kmers, no kmers were found with their FDR <0.01 in m-human1. However, 14/35 (kmer = 3), 51/115 (kmer = 4), 158/209 (kmer = 5), and 513/526 (kmer = 6) kmers negatively correlated with the half-lives of mRNAs in m-human2 (FDR <0.01, S45–S48 Tables). For mRNA cDNA-based kmers, the results demonstrated that 2/3 (kmer = 3), 3/3 (kmer = 4), 2/2 (kmer = 5), and 4/4 (kmer = 6) kmers positively correlated with the half-lives of mRNAs in m-human1 (FDR <0.01). For m-human2, 19/42 (kmer = 3), 65/135 (kmer = 4), 162/259 (kmer = 5), and 554/580 (kmer = 6) kmers negatively correlated with the half-lives of mRNAs (FDR <0.01, S49–S52 Tables). Therefore, most kmers reduce mRNA stability.

For K2, we obtained similar results as those in K1. For example, 2/2 (kmer = 3), 22/22 (kmer = 4), 62/62 (kmer = 5), and 12/12 (kmer = 6) kmers positively correlated with the half-lives of lncRNAs in lnc-human1 (FDR <0.01, S53–56 Tables), and no kmers with their FDR <0.01 in lnc-human2. For m-human1 (CDS-based), only 1/1 (kmer = 5) kmer promote mRNA stability (FDR <0.01). For m-human2 (CDS-based), 16/37 (kmer = 3), 40/55 (kmer = 4), 139/140 (kmer = 5), and 149/151 (kmer = 6) kmers negatively correlated with half-lives of mRNAs (S57–S60 Tables). Additionally, among the 37 codons (FDR <0.01, kmer = 3), 21 codons (optimal codons) positively and 16 codons (non-optimal codons) negatively correlated with the half-lives of mRNAs. The detailed results are displayed in Fig 8. Further analysis showed a significant positive correlation between the half-lives of mRNAs and the contents of optimal codons ($P$ = 1.40E-54), and a negative correlation for non-optimal codons ($P$ = 1.32E-162). In sum, the results from kmer-based analysis were basically the same as those from protein-RNA or miRNA-RNA interaction analysis.

## Comprehensive analysis of lncRNA stability

Previous sections demonstrated that the half-lives of both lncRNAs from lnc-human1 and mRNAs from m-human2 are associated with transcripts lengths, GC contents, transcripts secondary structures, miRNA−RNA interactions, and protein−RNA interactions. In fact, associations also exist among different factors. For example, in lnc-human1, the correlation coefficients between lncRNA lengths and their secondary structure free energy reached −9.73E−1. To display the relationship among these factors and half-lives clearly, we constructed half-

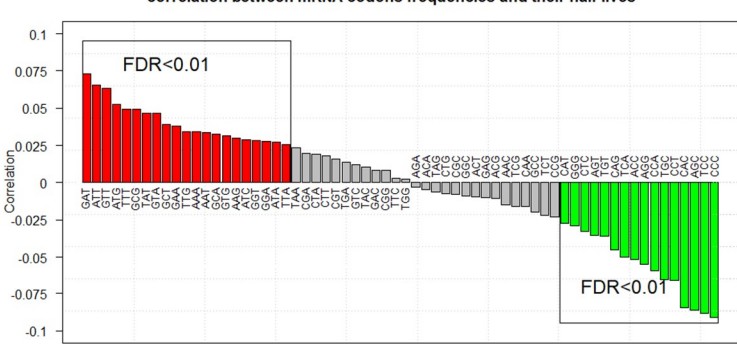

**Fig 8. Spearman correlation coefficients between the half-lives of mRNAs and their codon contents are displayed, in which the red bars stand for the positive correlation of the 21 codons with their FDR values less than 1.00E-2 and the green bars stand for the negative correlation of the 16 codons with their FDR values less than 1.00E-2.**

life regulation networks for both lncRNAs in lnc-human1 and mRNAs in m-human2 using Spearman correlation analysis (Fig 9), in which the edge would be generated if the FDR value from the related two nodes is less than 0.01. The half-life of lncRNAs was found to be regulated by less number of factors compared with mRNA half-life regulation. Additionally, although the positive or negative correlation among five factors in lnc-population1 was consistent with the counterpart in m-human2, all these five factors play a contrary role in regulating half-lives. Therefore, the degradation mechanisms for both lncRNAs in lnc-human1 and mRNAs in m-human2 should be different.

Finally, we intended to investigate the quantitative relationship between the half-lives of lncRNAs (mRNAs) and the related factors and their combinations using regression analysis. To this end, we first extracted as many related features as possible. For lncRNAs, the total number of features is 10,418, which included 5460 kmer compositions (kmer = 1:6), 2297 protein-binding RNA motif compositions and 1 total composition, 2656 miRNA-binding site compositions and 1 total composition, and 3 basic properties for the full-length sequence (RNA secondary structure, length, and GC content). A total of 10,425 features are present for mRNAs, which included the similar features mentioned earlier and the features from 5' end UTR, CDS, and 3' end UTR. Then we calculated the correlations between the half-lives of lncRNAs (mRNAs) and each feature, and chose the features with $P$ values (FDR for mRNAs) less than 0.1, 0.05, and 0.01 for developing prediction models. The numbers of selected features for lncRNAs (mRNAs) are 1001 (1495), 2075 (1695), and 2918 (1807). The results indicated that the adjusted $R^2$ values for lnc-human (m-human) were 2.30E-2 (8.63E-2), 1.91E-2 (8.48E-2), and 1.35E-2 (8.26E-2) using the features with their $P$ values (FDR for mRNAs) less than 0.1, 0.05, and 0.01, respectively. Therefore, the linear regression models could not reflect the quantitative relationship between the half-lives of lncRNAs (mRNAs) and the related factors and their combinations. We decided to apply deep learning-based regression to explore the nonlinear relationship. The packages keras and tensorflow were used to develop a regression model.

In our model, except the input and output layers, six hidden layers are present with the number of neurons as 500, 250, 100, 50, 30 and 10, respectively. The activation function "relu" was used. During the training stage, we set up epochs = 100, batch_size = 1000, and validation_split = 0.2. The results from lnc-human demonstrated that the Spearman correlation coefficients between the real half-lives and predicted half-lives are 8.07E-1 ($P$ = 0.00), 8.00E-1 ($P$ = 0.00), and 7.58E-1 ($P$ = 0.00) using the features with their $P$ values less than 0.1, 0.05, and

**half-life regulation network for lnc-human1**   **half-life regulation network for m-human2**

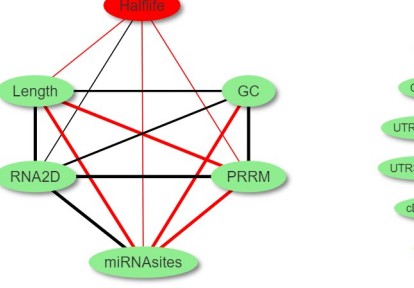 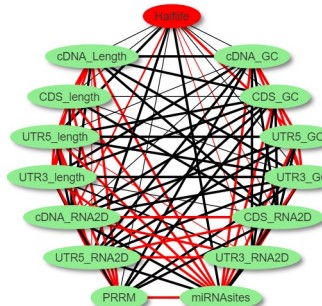

**Fig 9. Half-life regulation networks for lnc-human1 and m-human2; red lines stand for positive correlation and black lines for negative correlation.** The thick, medium, and thin lines represent the strong ($P < 1.0E-33$), medium (p∈[1.0E−33, 1.0E−26]), and weak (p∈(1.0E−26, 1.0E−5)) correlation, respectively. The meaning of the words Half-life, Length (_length), GC (_GC), and RNA2D (_RNA2D) in the lncRNA (mRNA) regulation network are lncRNA (mRNA) half-lives, length, GC contents, and RNA secondary structures. PRRM and miRNAsites are the total number of protein recognition motifs and miRNA-binding sites, respectively, in lncRNAs or mRNAs.

0.01, respectively. For m-human, the Spearman correlation coefficient is 7.37E-1 ($P = 0.00$), 7.31E-1 ($P = 0.00$), and 7.27E-1 ($P = 0.00$) using the features with their FDR values less than 0.1, 0.05, and 0.01, respectively. Therefore, there indeed exist a nonlinear relationship between the the half-lives of lncRNAs (mRNAs) and the related factors and their combinations. However, for lnc-human, the correlation coefficient from fivefold cross-validation isonly 9.10E-2 ($P = 1.19E-36$), 8.26E-2 ($P = 1.14E-49$), and 6.62E-2 ($P = 1.67E-32$) using the features with their $P$ values less than 0.1, 0.05, and 0.01, respectively. For m-human, the correlation coefficient from fivefold cross-validation is only 2.07E-1 ($P = 1.13E-219$), 2.05E-1 ($P = 1.87E-215$), and 2.03E-1 ($P = 6.32E-212$) using the features with their FDR values less than 0.1, 0.05, and 0.01, respectively. Additionally, similar results were obtained for lnc-human1 and m-human2. We also tried other network parameters such as different numbers of layers and different batch_sizes. The results from fivefold cross-validation are basically the same. Therefore, further effort is needed to adjust the network structure or incorporate other new features to obtain a better prediction model.

## Conclusions

In this study, RNA-seq and random sampling strategy were applied to calculate the half-lives of transcripts, the half-life datasets in human A549 cell lines were obtained, which included 33,285 lncRNAs, 24,710 mRNAs, nucleus-specific 5758 lncRNAs and 13,093 mRNAs, cytoplasm-specific 1515 lncRNAs and 9833 mRNAs, and nucleus and cytoplasm-common 491 lncRNAs and 2496 mRNAs. Then we systematically investigated the relationship between the half-lives of lncRNAs (mRNAs) and related factors such as transcript lengths, GC contents, RNA secondary structures, miRNA-RNA interactions, protein-RNA interactions, cellular location and kmer compositions, and obtained the following observations.

First, it is necessary to divide the lncRNAs (mRNAs) into lnc-human1 (m-human1) with only one exon and lnc-human2 (m-human2) with more than one exon in stability analysis. For example, in lnc-human1, a positive correlation existed between half-lives of lncRNAs and their lengths ($P = 1.90E-19$), or miRNA-binding ability ($P = 9.16E-13$), or protein-binding ability ($P = 5.74E-19$), and a negative correlation existed with RNA secondary structure free energy ($P = 8.24E-19$). However, no similar relationship was found in lnc-human2. Therefore, the degradation behavior of lnc-human1 and lnc-human2 is different. A similar situation exists in m-human1 and m-human2. Here we also want to point out that the number of mRNAs in m-human1 is only 474. A large number of mRNAs in m-human1 is necessary to check the reliability of the conclusions.

Second, according to the kmer composition-based analysis for the first mode K1, we found that the kmer compositions mainly influence the stabilities of both lncRNAs from lnc-human1 and mRNAs from m-human2. For lnc-human1, all kmers with their FDR<0.01 positively correlated with the half-lives of lncRNAs, which includes 1, 16, 41, and 158 kmers for kmer = 3, 4, 5, and 6, respectively. For lnc-human2, only six kmers negatively correlated with the half-lives of lncRNAs. For m-human2, with the increase in kmer length, the ratios of the number of motifs negatively correlated with the stability and the number of motifs positively correlated with the stability become larger. For example, the CDS-based ratio for kmer = 3, 4, 5, and 6 is 0.67 (14/21), 0.80 (51/64), 3.10 (158/51), and 39.46 (513/13), respectively. The cDNA-based ratio for kmer = 3, 4, 5, and 6 is 0.83 (19/23), 0.89 (62/70), 1.57 (154/98), and 21.84 (546/25), respectively. For m-human1, however, only 2/3, 3/3, 2/2, and 4/4 kmers positively correlated with the half-lives for kmer = 3, 4, 5, and 6, respectively. In view that the kmer compositions mentioned earlier are calculated without any constraints, we deduced that the kmer (motif)-based interactions between lncRNA (mRNA) and protein or RNA would have the same conclusions. In fact, for

miRNA-RNA interaction-based RNA stability analysis, 32/37 miRNAs positively correlated with lncRNA stability from lnc-human1, 449/449 miRNAs negatively correlated with mRNA stability from m-human2, and no miRNAs correlated with lncRNA stability from lnc-human2 and mRNAs from m-human1. To further support the aforementioned results, we also considered the relationship between methylation and lncRNA (mRNA) stability.

According to the report [49], methyltransferase modified mRNA sequences by identifying RNA motif RRACH (R = G or A; H = A, C, or U), which is a motif-based modification. Methylation is assumed to increase lncRNA stability from lnc-human1 and decrease mRNA stability from m-human2. This is validated by calculating the occurrence numbers of the motif in each lncRNA (mRNA) sequence, and Spearman correlation between the half-lives of lncRNAs (mRNAs) and their motif contents. The results demonstrated a significant positive correlation for lnc-human1 ($P$ = 3.47E-14), and no correlation for lnc-human2 ($P$ = 1.42E-2) or whole population lnc-human ($P$ = 5.59E-2). For mRNAs, a significant negative correlation was found for m-human2 ($P$ = 1.15E-38) or whole population m-human ($P$ = 1.70E-34), and no correlation for m-human1 ($P$ = 1.12E-2).

Third, we found that cellular location (nucleus and cytoplasm) did not significantly influence the stability of both lncRNAs and mRNAs. According to the expression profiles in the nucleus and the cytoplasm at 0, 6, 12, and 24 h, two strategies, enrichment analysis and direct half-life calculation, were applied to study the influence of cellular location (nucleus and cytoplasm) on lncRNA (mRNA) stability. The results from enrichment analysis showed that the average half-life of the lncRNA (mRNA) population in the nucleus is indeed less than that of the lncRNA (mRNA) population in the cytoplasm, which is consistent with the Clark conclusions on lncRNA half-life study in mice [19]. In their study, Clark et al. obtained the conclusions using the half-life dataset of 105 lncRNAs enriched in the nucleus and 22 lncRNAs enriched in the cytoplasm. This result was also validated by the report [21]. However, according to the half-lives of the common 491 lncRNAs and 2496 mRNAs in the nucleus and the cytoplasm, which were from the direct half-life calculation in our study, no significant difference was found. Therefore, cellular location do not have a significant influence on these lncRNAs and mRNAs. We deduced that the lower average half-life of the lncRNA (mRNA) population in the nucleus is caused by the transcripts themselves or other unknown factors rather than cellular location. The initial analysis showed that GC-type kmers such as CG, CGG, and CCG usually had less content in nucleic-specific lncRNAs (mRNAs) than those in cytoplasm-specific lncRNAs (mRNAs). These kmers might be the parts of potential target regions for protein-RNA or RNA-RNA interactions, which rendered the lncRNAs (mRNAs) less stable in the nucleus than in the cytoplasm.

Fourth, besides the close relationship between mRNA stability and codons usage, the secondary structures of cDNA, CDS, 5' end UTR, and 3' end UTR were also found to have a close relationship with their half-lives. Specially, stable 5' UTR or unstable 3' UTR secondary structure promote mRNA stability. Therefore, a comprehensive analysis for mRNA stability is necessary.

Finally, through quantitative analysis between the half-lives of lncRNAs (mRNAs) and various factors, we found a nonlinear relationship between the half-lives of lncRNAs (mRNAs) and the related factors and their combinations. Further efforts are needed to develop an accurate quantitative prediction model for the half-lives of lncRNAs (mRNAs). In future, we will also pay attention to stability analysis of lncRNAs from lnc-human2 and mRNAs from m-human1.

## Methods

### Cell culture, transcription inhibition, and RNA purification

Human lung adenocarcinoma cells (A549, ATCC' CCL-185) were incubated at 37˚C and 5% $CO_2$ in a humidified atmosphere with DMEM medium (Sigma, 8119235) containing 10% fetal

bovine serum (PAN, ST30-3302) and antibiotics (100 U/mL of penicillin and 0.1 mg/mL of streptomycin). We added DMEM medium with 30 μg/mL actinomycin D (Sigma, A1410) to inhibit RNA transcription when cells reached 60%-70% confluency. Then we harvested cells in 0, 0.5, 1, 1.5, 2, 3, 4, 5, 6, and 8 h after inhibiting transcription. The total RNA was exacted and purified with an RNeasy Plus Mini kit (QIAGEN, 74134) as per manufacturer's instructions. All experiments were repeated three times. Finally, we got 30 samples for RNA-seq (see S1 Text).

## Purification of cytoplasm and nucleus RNA

Cell lysis reagent was used to separate the cytoplasm and the nucleus according to the literature [50], and cells were harvested in 0, 6, 12, and 24 h after inhibiting transcription using actino-mycin D. Subsequently, the cell membrane was lysed by lysate buffer and the nucleus and cyto-plasm were separated by differential centrifugation. Finally, the nucleus RNA was extracted using Trizol (Sigma, 93289) and cytoplasm RNA using TRI Reagent (Sigma, T3934) (see S1 Text and S1 Fig).

## RNA-seq and data processing

For RNA sequencing, stranded cDNA libraries of 30 samples were generated using Illumina Stranded Total RNA Prep (illumina, 20040525) and sequenced on the Illumina HiSeq4000 by IgeneCode Biotech (Beijing, China). After obtaining the raw sequencing datasets, the low-quality reads were removed, which included reads with the adaptors, reads with the ratio of bases $N \geq 5\%$, and reads with the ratio of low-quality bases (quality score $\leq 10$) $\geq 20\%$. Subse-quently, the software SOAP2 was used to remove the reads from rRNAs (-m 0 -x 1000 -s 40 -l 32 -v 5 -r 2 -p 3) [51]. After filtering, about 80 million reads were obtained for each sample. Finally, the software HISAT (–phred33 –sensitive–no-discordant–no-mixed -I 1 -X 1000 –rna-strandness RF) [52] was used to map the remaining clean reads onto the reference genomes, StringTie(-f 0.3 -j 3 -c 5 -g 100 -s 10000 -p 8) to reconstruct transcripts [53], and Ballgown (-B) to calculate transcript expression levels [54]. Through mapping results, we obtained the expression profiles of both lncRNAs from NONCODE [31] and mRNA from the refseq databases [55], then we obtained the expression profiles of both lncRNAs and mRNAs in FPKM. Considering that the experiments were performed in triplicate, each transcript had three time series of expression data, which were marked as $A = A_1A_2 \ldots A_{10}$, $B = B_1B_2 \ldots B_{10}$, and $C = C_1C_2 \ldots C_{10}$, respectively. The subscripts of the A, B, and C represented the time points 0.0, 0.5, 1.0, 1.5, 2.0, 2.5, 3.0, 5.0, 6.0, and 8.0 h, sequentially. For each time point, pair-wise Pearson correlation coefficients were also calculated among three replicates, and the min-imum coefficients were 0.73 ($P = 0.0$) for mRNA and 0.85 ($P = 0.0$) for lncRNA. Therefore, the experiments in the present study had good reproducibility. Finally, the expression profiles from three replicates were used for half-life calculation. The new transcripts were not consid-ered because of no systematic annotations for them.

## Calculation of half-lives of transcripts

According to the expression profiles of both lncRNAs and mRNAs in three time series *A*, *B*, and *C* (Fig 1 and see S1 and S2 Tables for detailed information), their half-lives were calculated using the following sampling strategy [1,21,23]: ① For a particular transcript *T*, a new time series data $R = R_1R_2 \ldots R_{10}$ was generated using *T*'s expression profile $A = A_1A_2 \ldots A_{10}$, $B = B_1B_2 \ldots B_{10}$, and $C = C_1C_2 \ldots C_{10}$, in which $R_i$ was randomly taken as one of the three val-ues $A_i$, $B_i$, and $C_i$ ($i = 1, 2, \ldots, 10$); ② The *T*'s half-life was calculated using the $R = R_1R_2 \ldots R_{10}$ by the function pk.calc.half.life in R package PKNCA; ③ Through repeating the steps ① and ② 1000 times, 1000 half-lives were obtained for the transcript *T*. ④ The average of half-lives

with their adjusted $R^2 \geq 0.7$ in step ③ was taken as the *T's* half-life. ⑤ Through repeating steps ① ~ ④ for each transcript, the half-life dataset was obtained for all transcripts. ⑥ Through repeating the steps ① ~ ⑤ 10 times, 10 half-life datasets were obtained. ⑦ The common transcripts in 10 simulations in step ⑥ were the final datasets, and the average half-life and associated 95% confidence interval for each transcript were calculated.

To demonstrate the robustness of the aforementioned sampling strategy, one-way ANOVA was applied to detect the differences in the half-lives of transcripts among 10 simulations. The results indicated no significant differences with *P* values as 1.00 for both lncRNAs and mRNAs.

## Biocomputation

Here all statistical calculations were completed in R language. For example, the function pk.calc.half.life in R package PKNCA was used to calculate the half-lives of transcripts with the following parameters: manually.selected.points = FALSE, conc.na = "drop", conc.blq = "drop", allow.tmax.in.half.life = TRUE, and check = TRUE, and the the half-lives of transcripts were selected with their adj.r.squared $\geq 0.7$ for further analysis. Additionally, Spearman correlation coefficient was used to describe the relationship between the two features, and the Kolmogorov-Smirnov test was used to detect the difference between the two populations. The related functions were cor.test and ks.test, respectively. Bonferroni correction was also used to calculate FDR values. The function lm was used to fit linear models. The package visNetwork was applied to display a half-life regulation network. The packages keras and tensorflow were used to develop a deep learning-based regression model.

## Bioinformatics analysis

The program RNAFOLD was used for predicting the RNA secondary structure (-d2—noLP—noClosingGU) [41], miRanda for predicting miRNA targets for both lncRNAs and mRNAs (-sc 140 -en -5 -scale 4 -strict -go -4 -ge -9 –quiet)[43], and PANTHER for GO annotation (http://geneontology.org/).

## LncRNA classification

LncRNA is classified into five categories using its position on the reference genome (Fig 10). Sense lncRNAs can be considered as transcript variants of protein-coding mRNAs, as they overlap with a known annotated gene on the same genomic strand. Antisense lncRNAs are RNA molecules transcribed from the antisense strand and overlap in part with well-defined spliced sense or intronless sense RNAs. Intergenic lncRNAs are long non-coding RNAs that locate between annotated protein-coding genes, and are at least 1 kb away from the nearest protein-coding genes. Intronic lncRNAs are RNA molecules that overlap with the intron of annotated coding genes in either sense or antisense orientation. Bidirectional lncRNAs are oriented head to head with a protein-coding gene within 1 kb. This study considered only the first four categories provided by the NONCODE database[56].

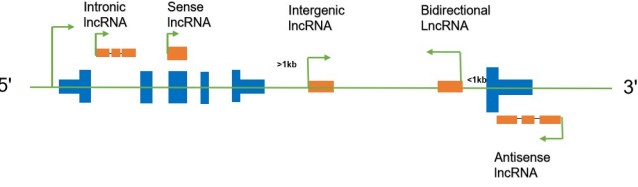

**Fig 10. Five categories of lncRNAs.**

## Supporting information

**S1 Table. LncRNA expression profiles.**
(CSV)

**S2 Table. mRNA expression profiles.**
(CSV)

**S3 Table. Half-lives of lncRNAs.**
(CSV)

**S4 Table. Half-lives of mRNAs.**
(CSV)

**S5 Table. LncRNA exon list.**
(CSV)

**S6 Table. mRNA exon list.**
(CSV)

**S7 Table. LncRNA classification.**
(CSV)

**S8 Table. Secondary structures of full-length lncRNAs.**
(CSV)

**S9 Table. Secondary structures of full-length mRNA cDNAs.**
(CSV)

**S10 Table. Secondary structures of full-length mRNA CDSs.**
(CSV)

**S11 Table. Secondary structures of lncRNA 5′ end fragments.**
(CSV)

**S12 Table. Secondary structures of lncRNA 3′ end fragments.**
(CSV)

**S13 Table. Secondary structures of mRNA 5′UTR fragments.**
(CSV)

**S14 Table. Secondary structures of mRNA 3′UTR fragments.**
(CSV)

**S15 Table. LncRNA expression profiles in nucleus and cytoplasm.**
(CSV)

**S16 Table. mRNA expression profiles in the nucleus and the cytoplasm.**
(CSV)

**S17 Table. Half-lives of nucleus-specific lncRNA.**
(CSV)

**S18 Table. Half-lives of nucleus-specific mRNAs.**
(CSV)

**S19 Table. Half-lives of cytoplasm-specific lncRNAs.**
(CSV)

**S20 Table. Half-lives of cytoplasm-specific mRNAs.**
(CSV)

**S21 Table. Half-life dataset of the nucleus and cytoplasm-common lncRNAs in the nucleus.**
(CSV)

**S22 Table. Half-life dataset of the nucleus and cytoplasm-common lncRNAs in the cytoplasm.**
(CSV)

**S23 Table. Half-life dataset of the nucleus and cytoplasm-common mRNAs in the nucleus.**
(CSV)

**S24 Table. Half-life dataset of the nucleus and cytoplasm-common mRNAs in the cytoplasm.**
(CSV)

**S25 Table. Prediction-based miRNA-lncRNA interaction list.**
(RAR)

**S26 Table. Role of miRNA-lncRNA interaction on lncRNA stability in lnc-human1.**
(CSV)

**S27 Table. Prediction-based miRNA-mRNA interaction list.**
(RAR)

**S28 Table. Role of miRNA-mRNA 5´UTRs interaction on mRNA stability in m-human2.**
(CSV)

**S29 Table. Role of miRNA-mRNA CDSs interaction on mRNA stability in m-human2.**
(CSV)

**S30 Table. Role of miRNA-mRNA 3´UTRs interaction on mRNA stability in m-human2.**
(CSV)

**S31 Table. Role of protein-lncRNA interaction on lncRNA stability in lnc-human1 (motif level).**
(CSV)

**S32 Table. Role of protein-lncRNA interaction on lncRNA stability in lnc-human1 (protein level).**
(CSV)

**S33 Table. Role of protein-mRNA 5´UTR interaction on mRNA stability in m-human2 (motif level).**
(CSV)

**S34 Table. Role of protein-mRNA 5´UTR interaction on mRNA stability in m-human2 (protein level).**
(CSV)

**S35 Table. Role of protein-mRNA CDS interaction on mRNA stability in m-human2 (motif level).**
(CSV)

**S36 Table. Role of protein-mRNA CDS interaction on mRNA stability in m-human2 (protein level).**
(CSV)

**S37 Table. Role of protein-mRNA 3´UTR interaction on mRNA stability in m-human2 (motif level).**
(CSV)

**S38 Table. Role of protein-mRNA 3´UTR interaction on mRNA stability in m-human2 (protein level).**
(CSV)

**S39 Table. kmer3-based lncRNA stability analysis in lnc-human1.**
(CSV)

**S40 Table. kmer4-based lncRNA stability analysis in lnc-human1.**
(CSV)

**S41 Table. kmer5-based lncRNA stability analysis in lnc-human1.**
(CSV)

**S42 Table. kmer6-based lncRNA stability analysis in lnc-human1.**
(CSV)

**S43 Table. kmer4-based lncRNA stability analysis in lnc-human2.**
(CSV)

**S44 Table. kmer5-based lncRNA stability analysis in lnc-human2.**
(CSV)

**S45 Table. kmer3-based mRNA stability analysis in m-human2 (CDS level).**
(CSV)

**S46 Table. kmer4-based mRNA stability analysis in m-human2 (CDS level).**
(CSV)

**S47 Table. kmer5-based mRNA stability analysis in m-human2 (CDS level).**
(CSV)

**S48 Table. kmer6-based mRNA stability analysis in m-human2 (CDS level).**
(CSV)

**S49 Table. kmer3-based mRNA stability analysis in m-human2 (cDNA level).**
(CSV)

**S50 Table. kmer4-based mRNA stability analysis in m-human2 (cDNA level).**
(CSV)

**S51 Table. kmer5-based mRNA stability analysis in m-human2 (cDNA level).**
(CSV)

**S52 Table. kmer6-based mRNA stability analysis in m-human2 (cDNA level).**
(CSV)

**S53 Table. Codonlike kmer3-based lncRNA stability analysis in lnc-human1.**
(CSV)

**S54 Table. Codonlike kmer4-based lncRNA stability analysis in lnc-human1.**
(CSV)

**S55 Table. Codonlike kmer5-based lncRNA stability analysis in lnc-human1.**
(CSV)

**S56 Table. Codonlike kmer6-based lncRNA stability analysis in lnc-human1.**
(CSV)

**S57 Table. Codonlike kmer3-based mRNA stability analysis in m-human2 (CDS level).**
(CSV)

**S58 Table. Codonlike kmer4-based mRNA stability analysis in m-human2 (CDS level).**
(CSV)

**S59 Table. Codonlike kmer5-based mRNA stability analysis in m-human2 (CDS level).**
(CSV)

**S60 Table. Codonlike kmer6-based mRNA stability analysis in m-human2 (CDS level).**
(CSV)

**S1 Text. Experimental operation details.**
(PDF)

**S1 Fig. qRT-PCR showing GAPDH enriched in cytoplasm and Malat1 enriched in nucleus, indicating that nucleus and cytoplasm were separated well.**
(TIF)

## Author Contributions

**Conceptualization:** Wuju Li, Xiaofei Zheng.

**Data curation:** Kaiwen Shi, Tao Liu, Wuju Li.

**Formal analysis:** Kaiwen Shi, Wuju Li.

**Funding acquisition:** Wuju Li, Xiaofei Zheng.

**Investigation:** Kaiwen Shi, Hanjiang Fu, Xiaofei Zheng.

**Methodology:** Kaiwen Shi, Hanjiang Fu, Xiaofei Zheng.

**Project administration:** Wuju Li, Xiaofei Zheng.

**Resources:** Hanjiang Fu, Wuju Li, Xiaofei Zheng.

**Software:** Kaiwen Shi, Tao Liu, Wuju Li.

**Supervision:** Wuju Li, Xiaofei Zheng.

**Validation:** Kaiwen Shi, Hanjiang Fu, Xiaofei Zheng.

**Visualization:** Kaiwen Shi, Tao Liu, Wuju Li, Xiaofei Zheng.

**Writing – original draft:** Kaiwen Shi, Wuju Li.

**Writing – review & editing:** Kaiwen Shi, Tao Liu, Hanjiang Fu, Wuju Li, Xiaofei Zheng.

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
