## [Decision Letter · Decision Letter 0]

18 Nov 2020

Dear Mr SHI,

Thank you very much for submitting your manuscript "Genome-wide analysis of lncRNA stability in human" for consideration at PLOS Computational Biology.

As with all papers reviewed by the journal, your manuscript was reviewed by members of the editorial board and by several independent reviewers. In light of the reviews (below this email), we would like to invite the resubmission of a significantly-revised version that takes into account the reviewers' comments.

One of the critical issues is with the way the manuscript is written, both scientifically and language-wise. A substantial revision is expected should you decide to resubmit the manuscript. 

We cannot make any decision about publication until we have seen the revised manuscript and your response to the reviewers' comments. Your revised manuscript is also likely to be sent to reviewers for further evaluation.

Sincerely,

Shi-Jie Chen

Associate Editor

PLOS Computational Biology

Florian Markowetz

Deputy Editor

PLOS Computational Biology

Reviewer's Responses to Questions

**Comments to the Authors:**

Reviewer #1: Abstract:

(i) use of technology/methods is missing.

(ii) Author claims “Our study established lncRNA and mRNA half-life regulation networks in human and shed new light on the degradation behaviors of both lncRNAs and mRNAs.”

Comment: But it’s not clear either authors want to base this claim on just one cell line which they used, and they want to claim that in all human cell lines this statement will hold true? Please clarify and write statements based only on the evidence that you have.

Material and methods:

(i) Please include the parameters used for each software used.

(ii) Through combining the mapping results and the annotations from 47 NONCODE[31]and refseq databases[55], we obtained the expression profiles of both lncRNAs and mRNAs in fpkm.

Comment: please elaborate the meaning of “Through combining” and write clearly i.e., why and how?

(iii) The packages keras and tensorflow were applied to develop deep learning-based regression model.

Comment: please write all the commends you used for the modelling in this section, prospectively, make a git-repo for the readers for all the commends, and libraries (R/python) for the results reproduction.

Results:

(i) Lines from 104-124, should be moved to “material and methods” section. They are not representing results. Results should start from 124-.

(ii) Line- 133, “The average 133 8(median) half-lives of lncRNAs and mRNAs are 3.96 hours (2.76 hours) and 1346.35 hours (4.18 hours), respectively.”

Comment: is it average or median?

General comments: Scientifically English must be written wisely and properly. Please consider carefully writing your paper, it will ensure better understanding and value to your hard work. In its present shape, the paper is very difficult to understand, especially the results and discussion section.

Reviewer #2: The authors provided a detailed profile of lncRNA and mRNA stabilities using RNASeq time series in human A549 cell line. RNA stability is an important issue, and this work provided good resource and some interesting results. The main problem is that the authors did not presented their data in an intuitive way. There are too many numbers and p values in the text, while data details are not shown. It’s better to show many of these data in figures. For example, correlation of half-lives and expression levels can be shown in a scatter plot. Also, the English language need to be substantially improved.

Other points:

1. Page 10, line 177, “It was also shown that lncRNAs have the same situation in mouse.” What about mRNAs in mouse or lncRNAs in thaliana?

2. The authors did not describe clearly how lncRNAs are classified (sense, intergenic, intronic and antisense).

3. Page 13, line 222, “For each class, the average half-life from lnc-human1 is always larger than that from lnc-human2.” The Results are not shown.

4. Page 15, line 252, the description of regions used for mRNA structure is not consensus with results in Figure 5. For lncRNAs, 5’ and 3’ UTRs make no sense. All regions are untranslated regions. What do the sizes of round dots mean in Figure 5? What about p values?

5. The half-life of nucleus and cytoplasm-specific lncRNAs (mRNAs) are not defined clearly. The half-life is calculated using time series, while nucleus and cytoplasm-specific lncRNAs (mRNAs) change over time. Is the half-life in nucleus calculated with nucleus only expression or both nucleus and cytoplasm expression? Are lncRNAs (mRNAs) transferring between nucleus and cytoplasm taken into account?

6. Page 28, it is curious that miRNAs promote lncRNA stability. What about expression of these miRNAs?

7. For protein-lncRNA interactions, are expressions of RNA binding proteins in the cell line taken into account?

8. In Figure 9, what about networks for lnc-human2 and m-human1?

9. Supporting tables are not available. The website http://ccb1.bmi.ac.cn:81/lncrnastability/ is down right now.

Reviewer #3: The authors have performed a comprehensive analysis of various factors on the half-lives of both mRNAs and lncRNAs. Interestingly, they identified factors associated with the stability of lncRNAs with 1 exon, but these factors lost relevance in case of stability of lncRNAs with more than one exons. Similarly, many factors were identified to be related to the stability of mRNAs with more than one exons, but not with mRNAs with one exon. Many factors like miRNA or RNA binding protein interaction motifs had opposite effects on the stability of lncRNAs and mRNAs. The associations were mostly identified using Spearman correlation coefficients. Finally, the authors tried to use regression models for predicting the half-lives of mRNAs and lncRNAs from the factors they identified to be associated with lncRNA or mRNA stability (through Spearman correlation). However, a linear regression was not found to be effective, and a deep-learning based regression model did not perform adequately in 5-fold cross-validation.

This work is definitely of interest because extensive analysis of a large number of factors related to the stability of lncRNAs and mRNAs. However, there remains some points to address:

1. The authors used Kolmogorov-Smirnov test for checking the difference between distributions of the half-lives of lncRNAs and mRNAs (nuclear vs cytoplasmic, or lncRNAs (mRNAs) with one exon vs more than one exon, or the different classes of lncRNAs). For further confidence in the results, the authors can perform t-tests or Wilcox tests to check if there are significant differences in mean (or median) half-lives between these categories. Visualizations using violin plots could reflect both the differences between medians along with the differences between the distributions.

2. For the final regression model, I wonder whether it could be effective to choose some of the most important parameters using LASSO or Elastic net models instead of using a very large number of parameters chosen based on Spearman's correlation coefficient.

Minor comments:

There are some sentence construction errors in the manuscript. Please check thoroughly.

**Have all data underlying the figures and results presented in the manuscript been provided?**

Reviewer #1: Yes

Reviewer #2: Yes

Reviewer #3: Yes

PLOS authors have the option to publish the peer review history of their article (what does this mean?). If published, this will include your full peer review and any attached files.

Reviewer #1: No

Reviewer #2: No

Reviewer #3: No
---

## [Decision Letter · Decision Letter 1]

26 Mar 2021

Dear Mr SHI,

We are pleased to inform you that your manuscript 'Genome-wide analysis of lncRNA stability in human' has been provisionally accepted for publication in PLOS Computational Biology.

You have the option to make the minor changes suggested by Reviewer #2.

Best regards,

Shi-Jie Chen

Associate Editor

PLOS Computational Biology

Florian Markowetz

Deputy Editor

PLOS Computational Biology

Reviewer's Responses to Questions

**Comments to the Authors:**

Reviewer #1: Although authors responded to the concerns but response is still not satisfactory.

Reviewer #2: For the scatter plots between half-lives and expression levels, the expressions can be log transferred to make them more clearly.

For Figure 5, lncRNA 5UTR and 3UTR can be changed to 5’ end and 3’ end.

**Have all data underlying the figures and results presented in the manuscript been provided?**

Reviewer #1: Yes

Reviewer #2: Yes

PLOS authors have the option to publish the peer review history of their article (what does this mean?). If published, this will include your full peer review and any attached files.

Reviewer #1: No

Reviewer #2: No

---

## [Editor Report · Acceptance letter]

8 Apr 2021

PCOMPBIOL-D-20-01867R1 

Genome-wide analysis of lncRNA stability in human

Dear Dr Li,

I am pleased to inform you that your manuscript has been formally accepted for publication in PLOS Computational Biology. Your manuscript is now with our production department and you will be notified of the publication date in due course.

With kind regards,

Alice Ellingham
